atmospheric science/environmental science/geophysics

shale gas, pore structure, fractal dimension, Knudsen number, diffusion, permeability

**Author for correspondence:**
Xijian Li
e-mail: 575914635@qq.com

Electronic supplementary material is available online https://doi.org/10.6084/m9.figshare.c.5418245.

# Fractal characteristics of shale pore structure and its influence on seepage flow

Shengwei Wang[1,3,4], Xijian Li[1,3,4], Haiteng Xue[1,3,4], Zhonghui Shen[5] and Liuyu Chen[2]

[1]Mining College, Guizhou University, Guiyang 550025, People's Republic of China
[2]Guizhou Electric Power Design and Research Institute Co., Ltd, Power Construction Corporation of China, Guiyang 550081, People's Republic of China
[3]Engineering Center for Safe Mining Technology Under Complex Geologic Condition, Guiyang 550025, People's Republic of China
[4]Institute of Gas Disaster Prevention and Coalbed Methane Development of Guizhou University, Guiyang 550025, People's Republic of China
[5]State Key Laboratory for the Coal Mine Disaster Dynamics and Control, Chongqing University, Chongqing 400044, People's Republic of China

SW, 0000-0001-6178-2905; XL, 0000-0003-4160-4874;
HX, 0000-0002-1434-0212; ZS, 0000-0001-8009-9167;
LC, 0000-0002-1895-6946

The migration law of shale gas has a significant influence on the seepage characteristics of shale, and the flow of the gas is closely related to the pore structure. To explore the influence of shale pore parameters on permeability in different diffusion zones, the pore structure of the shale in the Niutitang Formation in Guizhou, China, was analysed based on liquid nitrogen adsorption experiments and nuclear magnetic resonance experiments. The relationship among fractal dimension, organic carbon content (TOC) and BET-specific surface area was analysed based on the fractal dimension of shale pores calculated using the Frenkel–Halsey–Hill model. Shale permeability was calculated using the Knudsen number ($Kn$) and permeability equation, and the influence of the fractal dimension and porosity in different diffusion zones on shale permeability was analysed. Previous studies have shown that: (i) the pores of shale in the Niutitang Formation, Guizhou are mainly distributed within 1–100 nm, with a small total pore volume per unit mass, average pore diameter, large BET specific surface area and porosity; (ii) fractal dimension has a negative correlation with average pore diameter and TOC content and a quadratic relationship with BET specific surface area; and (iii) permeability has a positive correlation with $Kn$, porosity and fractal dimension. In the transitional diffusion zone, fractal dimension and porosity have a significant impact on permeability. In the Knudsen diffusion zone, porosity has no obvious effect on permeability. The methodologies and results presented will enable more accurate characterization

of the complexity of pore structures of porous media and allow further understanding of the seepage law of shale gas.

# 1. Introduction

Shale is a typical dense porous medium with complex pore structures ranging in size from nanometres to micrometres [1]. As the mean free path of shale gas molecules is approximately $10^{-7}$, gas exhibits a different flow phenomenon in shale at different scales [2]. It follows a seepage process but also shows nonlinear flow such as diffusion and slip [3], which complicates the flow mechanism. Therefore, studying the mechanism of gas percolation in multi-scale pore structures is of great significance to the field of shale gas development.

Currently, international research groups have conducted considerable research on multi-scale seepage theory. Zheng et al. [4] established a fractal model of the gas diffusion coefficient in porous media using the fractal theory, and Zhang [5] established a porous membrane permeability model based on the fractal theory. Beskok & Karniadarkis [6] studied the flow of rarefied gases in channels, pipes and smooth surface pipes in the Knudsen diffusion range, obtained a general scale of the velocity profile and established a unified model using this scale. Yu & Cheng [7] established a fractal permeability model for double-dispersed porous media. Roy et al. [8] established a micro-scale flow model based on two-dimensional finite elements by simulating the flow of gas in microchannels and nanopores. Cai et al. [9] considered the fractal characteristics of porous media, derived the fractal characteristics based on porous media and proposed an analytical expression that characterizes the spontaneous co-current seepage process of wetting fluid into gas-saturated porous media; the model was used to predict the fluid absorption. The weight was in good agreement with the experimental data. Nan et al. [3] simulated the flow of methane in organic nanopores under shale reservoir conditions and used non-equilibrium molecular dynamics to simulate the transportation behaviour of nano-constrained methane molecules. They also studied the influence of pore size and pressure on slip length, providing a theoretical basis for the development of shale gas. Yu et al. [2] established a multi-scale lattice Boltzmann (LB) model considering the adsorption effect, simulated shale gas migration in micro-nanopores and studied the multi-scale migration mechanism of shale gas in microcracks. Wang et al. [10] used measured data to establish a model to determine the inherent permeability and estimated the permeability of the dense porous medium during the gas mass transfer process by considering the gas surface adsorption and slippage effects. Li et al. [11] analysed the correlation between the gas slip factor and pore anisotropy and Knudsen number ($Kn$) by establishing a two-dimensional beam model, combining the Bosanquet-type viscosity model and the LB model. Li et al. [12] derived the expression for effective gas permeability based on the microfluidic model and the fractal capillary model, considering the Klinkenberg effect. Although different scholars have studied multi-scale porous media gas seepage through experiments and mathematical models, relatively few studies exist on the effect of pore parameters in varying diffusion zones on shale permeability.

In this study, shale samples from the Niutitang Formation in Guizhou Province, China were selected and analysed via low-temperature liquid nitrogen adsorption experiments and nuclear magnetic resonance (NMR) experiments; the fractal characteristics of the pore structure of the Niutitang Formation were analysed based on the fractal dimension. According to the permeability equation of multi-scale porous media, the influence of the fractal dimension and porosity in different diffusion zones on permeability is further discussed to clarify the seepage law of shale and provide theoretical guidance for shale gas development in Guizhou.

# 2. Material and methods

## 2.1. Sample collection

The six samples collected for the experiment were obtained from Fenggang Fengcan 1 well, Cengong Tianma 1 well, and Dafang 1 well, all of which belong to the Lower Cambrian Niutitang Formation shale. The shale sample from well Fengcan 1 was at a depth of approximately 2500 m, that from well Tianma 1 at a depth of approximately 1500 m and that from well Dafang 1 at a depth of approximately 1000 m. A map of the study area location is shown in figure 1.

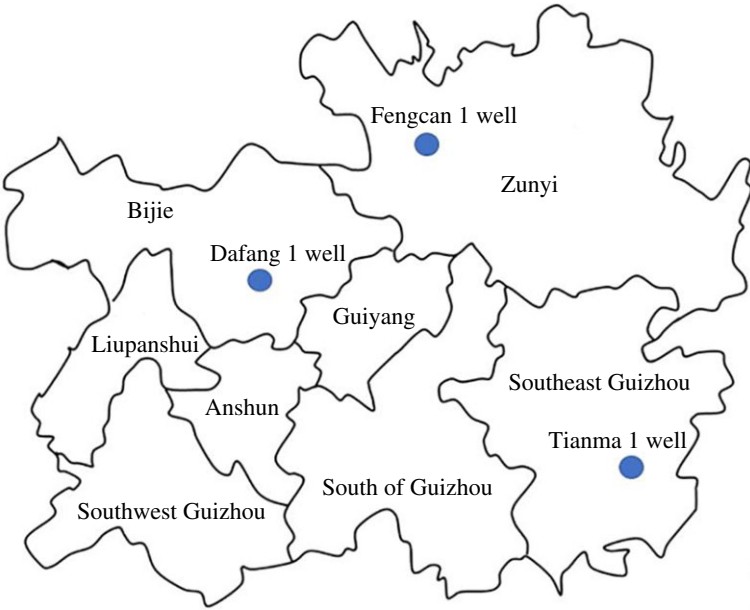

**Figure 1.** Location of the study area.

## 2.2. Experimental programme

(i) Organic carbon content determination, vitrine reflectivity and X-ray diffraction determinations were completed in the Guizhou Coal Field Geology Bureau Laboratory. The test basis for the determination of organic carbon content was GB/T19145-2003 that for the determination of vitrinite reflectance was SY/T5124-2012, and that for X-ray diffraction determination was SY/T5163-2010.

(ii) The liquid nitrogen adsorption experiment was determined using a 3H-2000PS1/2 static volumetric analyser. The measurement basis was GB/T19587-2004, and the pore size test range was 0.35–400 nm. The experimental sample processing steps are detailed in previous literature [13].

(iii) The NM12 type NMR analyser of the Numai Company was used for the NMR experiment.

# 3. Experimental results and analysis

## 3.1. Mineral composition analysis

Table 1 shows that the total organic carbon (TOC) content of shale was in the range of 3.94–7.50%, with an average value of 5.25%, and the overall organic matter content was relatively high [14]. The distribution range of $R_o$ was 1.685–3.112%, with an average of 2.36%, which corresponds to the stage of mature gas generation and early dry gas generation, under good shale gas reservoir effects and gas generation conditions [15]. The shale mineral composition of the Niutitang Formation mainly includes quartz 27.4–72.8% and clay minerals 8.9–42.6%; the clay minerals were mainly illite. There was also a small amount of albite 5.3–23.9% and pyrite 4.6–14.7%. The content of brittle minerals such as quartz was relatively high.

## 3.2. Nuclear magnetic resonance experiment

NMR spectroscopy can quantitatively characterize the pore state of shale and is widely used to test the pore structure and physical properties of shale [16]. As the transverse relaxation time ($T_2$) of the nuclear magnetic signal is proportional to the pore size, the shale pore size distribution can be calculated using the following equation [17]:

$$\frac{1}{T_2} = \rho_2 \left(\frac{S}{V}\right)_{\text{pore}}. \tag{3.1}$$

**Table 1.** X-ray diffraction analysis results of black shale. TOC: total organic carbon.

| shale samples | $R_o$ (%) | TOC (%) | mineral type and content (%) | | | | | | | clay mineral content fraction (%) | relative content of clay minerals (%) | | | | |
|---|---|---|---|---|---|---|---|---|---|---|---|---|---|---|---|
| | | | quartz | potash feldspar | albite | calcite | dolomite | pyrite | | | imon mixed layer | illite | kaolinite | chlorite |
| FC-1 | 2.270 | 3.94 | 27.4 | — | 18.4 | — | 5.8 | 5.8 | 42.6 | — | 100 | — | — |
| FC-2 | 2.260 | 5.01 | 47.2 | 2.8 | 23.9 | — | — | 9.1 | 17 | — | 100 | — | — |
| TM-2 | 3.112 | 5.05 | 38.6 | 1.8 | 12.7 | — | 4.3 | 14.7 | 27.9 | — | 99 | 1 | — |
| TM-3 | 1.352 | 4.25 | 72.8 | 1.0 | 5.3 | 4.8 | 2.6 | 4.6 | 8.9 | — | 100 | — | — |
| DF-2 | 2.444 | 7.50 | 39.6 | — | 15.2 | — | 7.7 | 9.3 | 28.2 | 8.9 | 81 | 8.9 | 1.2 |
| DF-3 | 1.873 | 4.02 | 46.9 | — | 21.9 | — | — | 9.4 | 21.8 | — | 89 | 7.6 | 4.4 |

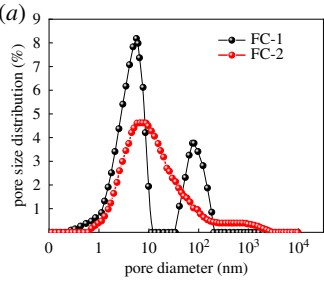 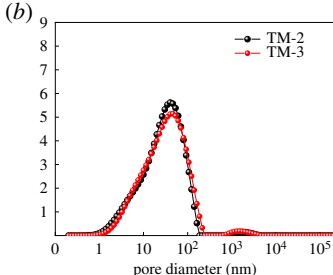 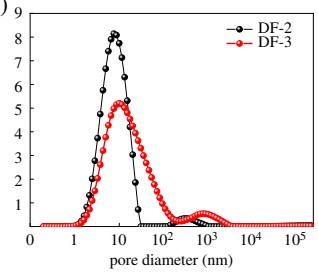

**Figure 2.** Shale pore size distribution.

**Table 2.** Porosity NMR test results.

| sample | saturated porosity (%) | centrifugal porosity (%) |
|---|---|---|
| FC-1 | 1.812 | 1.576 |
| FC-2 | 1.218 | 1.206 |
| TM-2 | 2.103 | 2.063 |
| TM-3 | 1.989 | 1.908 |
| DF-2 | 0.962 | 0.831 |
| DF-3 | 0.637 | 0.589 |

NMR experiments were performed on the shale samples, and the porosity (table 2) and pore size distribution (figure 2) of the experimental samples after saturation and centrifugation were obtained. The results in table 2 show that the saturated porosity distribution range was 0.637–2.103%, with an average of 2.22%, the centrifugal porosity distribution range was 0.589–2.063%, with an average of 1.36%, and the saturated porosity was approximately 1.6 times higher than the centrifugal porosity. This is because the movable water was partly separated after centrifugation, leaving behind the remaining part of the bound water. Due to the existence of some pores in the shale, water is bound in the pores and cannot be removed by centrifugation, resulting in a decrease in porosity. Therefore, the porosity after centrifugation was less than the saturated porosity.

It can be seen from figure 2 that the pore distribution of FC-1 presented a bimodal shape, the first peak corresponded to a pore size of approximately 1–10 nm, the second peak pore size was approximately 100 nm, and the pore distribution of FC-2 presented large left and a small right tendency. With a bimodal structure, the pore size distribution was mainly concentrated between 1 and 1000 nm, and the peak value of the pore size distribution was 8–10 nm. The pore size distributions of TM-2 and TM-3 were discontinuous bimodal structures with large left and small right peaks, and the right peak was relatively small. The pore size distribution of TM-2 was mainly concentrated in the range of 1–160 nm, the pore size distribution of TM-3 was mainly concentrated in the range of 1–220 nm and the pore size distribution peak was 50–60 nm. DF-2 and DF-3 were bimodal; the pore size distribution corresponding to the first peak of DF-2 was 1–30 nm, and the pore size distribution corresponding to the second peak was 100–1000 nm. The pore size distribution was 1–200 nm, the pore size distribution of the second peak was 300–5000 nm, and the pore size distribution peak was 10 nm. In summary, the shale pores of the Niutitang Formation in Guizhou were widely distributed in the range of 1–10 nm, indicating that the shale pore structure in this area is dominated by micropores.

## 3.3. Low-temperature liquid nitrogen adsorption experiment

The test results of low-temperature liquid nitrogen adsorption/desorption are shown in table 3, and the related curves are depicted in figure 3. Table 3 shows that the BET specific surface area of the six groups of shale samples was distributed from 7.2327 to 20.652 $m^2\,g^{-1}$, with an average of 13.73 $m^2\,g^{-1}$. The pore diameter ranged from 3.8732 to 6.3089 nm, with an average of 4.6 nm. Unit mass the total pore volume was distributed from 0.004209 to 0.014542 $cm^3\,g^{-1}$. The total pore volume per unit mass of the shale samples was small, the BET specific surface area was large and the average pore diameter was small,

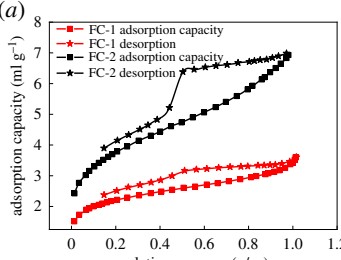
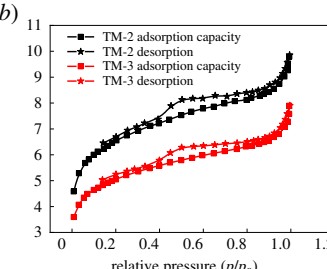
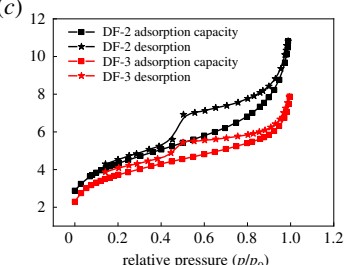

**Figure 3.** Low-temperature liquid nitrogen adsorption/desorption curve of the experimental sample (note: $p$ is the equilibrium pressure; $p_0$ is the gas adsorption reaching saturation vapour pressure).

**Table 3.** Test results of low-temperature liquid nitrogen adsorption/desorption experiment.

| sample | sampling depth (m) | BET specific surface area ($m^2 g^{-1}$) | average pore size (nm) | total pore volume per unit mass ($cm^3$) |
|---|---|---|---|---|
| FC-1 | 2461.11 | 12.7904 | 4.2448 | 0.009794 |
| FC-2 | 2518.99 | 7.2327 | 4.0125 | 0.004209 |
| TM-2 | 1487.63 | 20.652 | 3.8732 | 0.010062 |
| TM-3 | 1459.22 | 15.4600 | 4.1239 | 0.007937 |
| DF-2 | 996.14 | 14.1255 | 6.3089 | 0.014542 |
| DF-3 | 1021.27 | 12.1157 | 5.0693 | 0.009675 |

indicating that the Niutitang Formation shale micropores are generally developed. It can be seen from figure 3 that when the relative pressure $p/p_0$ was approximately 0.5, an obvious inflection point appeared, indicating adsorption hysteresis. When the relative pressure was close to 1, the degree of adsorption increased rapidly. At this stage, mesopores and macropores of the shale pore surface underwent capillary agglomeration, and the adsorption/desorption curve showed an obvious 'lag' phenomenon; according to the IUPAC classification, [18] it belongs to the $H_2$ type. The hysteresis ring indicated that there were more amorphous pores such as 'ink bottle'-shaped pores in the shale pores of the Niutitang Formation in Guizhou [19]. This type of shale has poor pore connectivity and low reservoir permeability; however, the reservoir displays suitable gas storage conditions.

## 3.4. Calculation of pore fractal dimension

We calculated the fractal dimensions of six groups of shale samples based on the Frenkel–Halsey–Hill (FHH) fractal model. The expression is as follows [20]:

$$\ln\left(\frac{V}{V_0}\right) = A + B \ln\left[\ln\left(\frac{p_0}{p}\right)\right], \tag{3.2}$$

where $V_0$ is the volume of gas adsorbed by the monolayer ($cm^3 g^{-1}$); $V$ is the amount of gas adsorbed under $P$, $cm^3 g^{-1}$; $A$ is a constant; and $B$ is the slope of the fitted line, which is linearly related to the fractal dimension.

In general, the value of the fractal dimension $D$ is between 2 and 3. The closer the fractal dimension is to 2, the smoother the pore surface is; the closer the fractal dimension is to 3, the less smooth the pore surface is and the more complex the pore structure is. The relationship between $B$ and $D$ can be expressed as

$$D = B + 3. \tag{3.3}$$

The liquid nitrogen adsorption curve reveals that when the relative pressure $p/p_0 > 0.5$, the adsorption and desorption curves exhibited obvious hysteresis, indicating that the physical properties of shale reflected by different pressure sections were different. Yin *et al.*'s research [21] shows that when the relative pressure $p/p_0$ is 0–0.5, the fractal dimension reflects the unevenness of the pore

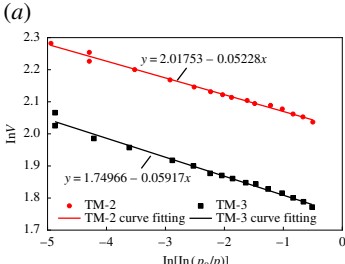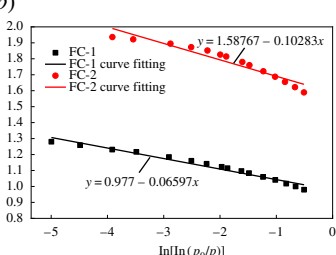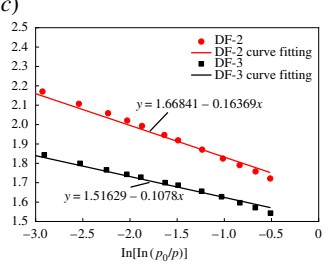

**Figure 4.** Adsorption capacity and relative pressure double log fitted curve.

**Table 4.** Calculated pore fractal dimension for each sample.

| experimental sample | linear equation | correlation coefficient $R^2$ | $B$ | $D = B + 3$ |
|---|---|---|---|---|
| FC-1 | $y = 0.977 - 0.06597x$ | 0.96795 | $-0.06597$ | 2.93 |
| FC-2 | $y = 1.58767 - 0.10283x$ | 0.91801 | $-0.10283$ | 2.89 |
| TM-2 | $y = 2.01753 - 0.05228x$ | 0.99225 | $-0.05228$ | 2.95 |
| TM-3 | $y = 1.74966 - 0.05917x$ | 0.98769 | $-0.05917$ | 2.94 |
| DF-2 | $y = 1.66841 - 0.16369x$ | 0.99142 | $-0.16369$ | 2.83 |
| DF-3 | $y = 1.51629 - 0.1078x$ | 0.99254 | $-0.10780$ | 2.90 |

surface; when the relative pressure $p/p_0$ is 0.5–1, the fractal dimension reflects the complexity of the pore structure. The seepage characteristics of shale are closely related to the shale pore structure. Therefore, the experimental data for a relative pressure of 0.5–1 was used, the graph was plotted according to equation (3.2), and the linear fit is shown in figure 4. The fractal dimension of the shale pores calculated using equation (3.3) is shown in table 4. It can be seen from table 4 that the fitting degrees of the samples in the study area were all > 0.9, and the $D$ values were all greater than 2.8, satisfying the fractal meaning of the pore system, and the pore structure is relatively complex.

## 3.5. Relationship between fractal dimension and pore structure

The fractal dimension can fully reflect the pore structure characteristics of shale [22]. The relationship among the pore fractal dimension and BET specific surface area, average pores and TOC content is shown in figure 5. Figure 5a demonstrates that the fractal dimension showed a quadratic relationship with the BET specific surface area, which indicates that with the increase of the fractal dimension, the BET specific surface area showed not a monotonic linear relationship, but rather a minimum point. This occurs because shale has different gas adsorption forces before and after the extreme point. Before the extreme point, the force between the gas molecules and shale is mainly the intermolecular force. Due to the collision between gas molecules, the gas molecules at the shale surface coverage decreases [19]. After the extreme point, the adsorption of gas molecules mainly relied on capillary cohesion, the interaction force between molecules gradually decreased, and the coverage of gas molecules on the shale surface increased. The fractal dimension had a negative correlation with the average pore diameter (figure 5b), indicating that the larger the fractal dimension was, the smaller the average pore diameter became. Figure 5c demonstrates that the fractal dimension of pores was negatively correlated with the TOC content, mainly because when the organic matter pores are not developed, although the organic matter content increases, the fractal dimension remains small.

# 4. Gas flow patterns in shale pores

## 4.1. Diffusion model of shale with different pore structure

Shale is a dense porous medium with highly developed micro-nano-scale pores and a wide range of pore diameters. The diffusion model of gas in shale pores is closely related to its pore size [21]. In this study,

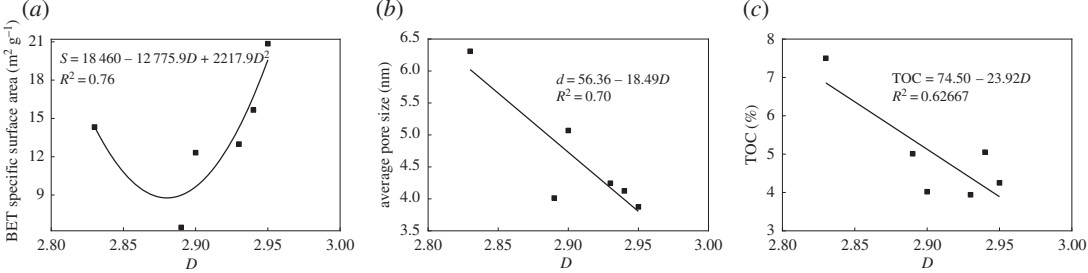

**Figure 5.** Fitting curve of pore fractal dimension and various influencing factors: (*a*) *D* and BET surface area, (*b*) *D* and average pore size and (*c*) *D* and TOC content.

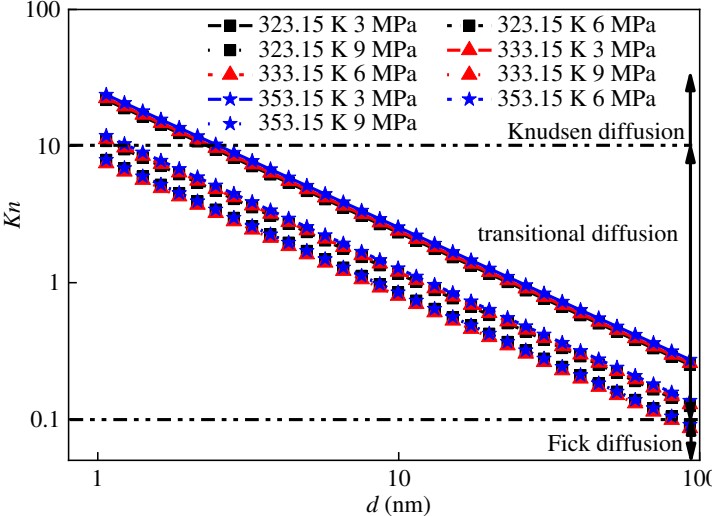

**Figure 6.** Pore gas diffusion mode under different pressure and temperature.

the Knudsen number was used to determine the gas diffusion model in the shale pores of the Niutitang Formation in Guizhou, and the expression is as follows [23]:

$$Kn = \frac{\lambda}{d}, \tag{4.1}$$

where *d* is the pore diameter (nm), *Kn* is the Knudsen number and $\lambda$ is the average molecular free path of the gas, nm,

$$\lambda = \frac{kT}{\sqrt{2}\pi d_0^2 p}, \tag{4.2}$$

where *k* is Boltzmann's constant, $1.38 \times 10^{-23}$ (J K$^{-1}$); $d_0$ is the gas molecule diameter, methane is 0.38 (nm); *T* is the absolute temperature, (K) and *p* is the gas pressure (MPa).

Combining equation (4.1) and equation (4.2), we can obtain

$$Kn = \frac{kT}{\sqrt{2}\pi d_0^2 pd}. \tag{4.3}$$

Previous studies [24] divide the gas diffusion modes into different ranges based on the value of *Kn*: (i) *Kn* > 10 which indicates Knudsen diffusion; (ii) 0.1 < *Kn* < 10 which indicates transition diffusion; (iii) 0.01 < *Kn* < 0.1 which indicates Fick diffusion; (iv) 0.001 < *Kn* < 0.01 which indicates slip flow; and (v) *Kn* < 0.001 which indicates Darcy flow. From the analysis in §3.2, it can be seen that the pore structure of shale comprises mainly micropores, with a small number of mesopores and macropores. To compare the difference of shale diffusion mode in different areas under the same pore size, we took the Fengcan 1 well as an example, according to the isotherm adsorption experimental conditions (temperatures of 50°C, 60°C and 80°C, the pressure of 0–10 MPa in literature [25]. According to (4.3), the *Kn* values of shale samples with pore diameters of 1–100 nm, temperatures of 323.15 K, 333.15 K and 353.15 K, and pressures of 3.0 MPa, 6.0 MPa and 9.0 MPa (figure 6) were calculated.

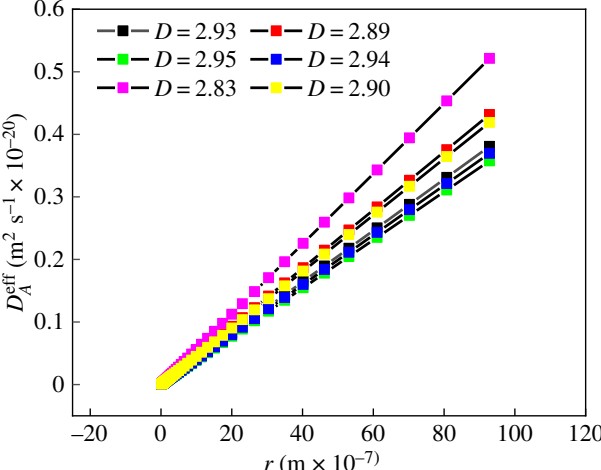

**Figure 7.** Variation of effective diffusion coefficient with aperture based on overdiffusion based on fractal dimension.

Figure 6 illustrates that when the pore size was in the range of 1–100 nm, the gas diffusion mode inside the shale of the Niutitang Formation in Guizhou was mainly transitional diffusion. Under the same pressure condition, the change in Knudsen number with pore size was positively correlated with temperature; the higher the temperature was, the greater the Knudsen number was but the lesser the influence was. Under the same temperature conditions, the change in Knudsen number with pore size was negatively correlated with pressure; the greater the pressure was, the smaller the Knudsen number was, and the more significant the rate of change was.

## 4.2. Diffusion coefficient calculation model

The Knudsen diffusion coefficient model can be expressed as [24]

$$C_k = \frac{2r}{3}\left(\frac{8RT}{\pi M}\right)^{0.5},\tag{4.4}$$

where $C_k$ is the Knudsen diffusion coefficient ($m^2\,s^{-1}$); $r$ is the pore radius (nm), $R$ is the gas constant (8.314 J mol$^{-1}$ K$^{-1}$); $T$ is the temperature (K) and $M$ is the molar mass of methane (kg mol$^{-1}$).

The Fick diffusion coefficient model can be expressed as [24]

$$C_f = \frac{kT}{6\pi\mu r_a},\tag{4.5}$$

where $C_f$ is the Knudsen diffusion coefficient, ($m^2\,s^{-1}$); $r_a$ is the molecular radius (nm) and $\mu$ is aerodynamic viscosity (Pa s).

The transition coefficient diffusion model can be expressed as [24]

$$C_{\text{transition}} = (C_k^{-1} + C_f^{-1})^{-1},\tag{4.6}$$

where $C_{\text{transition}}$ is transition diffusion coefficient, ($m^2\,s^{-1}$).

From equations (4.4)–(4.6), it can be seen that the factors affecting gas diffusion were related to temperature and pore size; however, the influence of the shale pore structure on the diffusion model cannot be reflected. Schiefeestein *et al.*'s experimental research on activated carbon shows that the relationship between the effective diffusion coefficient and diffusion coefficient is as follows [26]:

$$D_A^{\text{eff}} = C\left(\frac{\lambda_{\min}}{\lambda_{\max}}\right)^{0.5D+4.5},\tag{4.7}$$

where $D_A^{\text{eff}}$ is effective diffusion coefficient, ($m^2\,s^{-1}$); $C$ is the diffusion coefficient, ($m^2\,s^{-1}$).

Equation (4.6) shows that under the same conditions, the transition diffusion coefficient is much smaller than the Knudsen and Fick diffusion coefficients. Therefore, this article only discusses the variation of the effective diffusion coefficient of Knudsen diffusion and Fick diffusion with the fractal dimension and aperture, as shown in figure 7 and table 5, which show:

**Table 5.** Effective diffusion coefficient of Fick diffusion under different fractal dimensions.

| fractal dimension | 2.95 | 2.94 | 2.93 | 2.90 | 2.89 | 2.83 |
|---|---|---|---|---|---|---|
| fick effective diffusion coefficient | $1.50 \times 24^{-10}$ | $1.55 \times 24^{-10}$ | $1.60 \times 24^{-10}$ | $1.76 \times 24^{-10}$ | $1.81 \times 24^{-10}$ | $2.20 \times 24^{-10}$ |

(i) The effective diffusion coefficient of Knudsen diffusion increased with the increase in the pore size and had a strong positive correlation. The main reason is that the increase in pore size leads to an increase in the pore structure space, which makes gas diffusion easier. The effective diffusion coefficient in the Fick diffusion zone is unrelated to the pore diameter change.

(ii) Under the same aperture conditions, either Fick diffusion or Knudsen diffusion, as the fractal dimension increases, the effective diffusion coefficient becomes smaller. This is because the fractal dimension is a characterization of the complexity of the pore structure of shale. The larger the fractal dimension, the more complex the pore structure is, and the resistance to gas diffusion will increase, making gas diffusion more difficult. However, the effective diffusion coefficient of Knudsen diffusion was much larger than that of Fick diffusion. Therefore, the selection of different diffusion modes has a great influence on the calculation of the diffusion coefficient and permeability.

# 5. Analysis of gas flow in shale pores and its influencing factors

## 5.1. Gas permeability

According to Darcy's Law:

$$Q = \frac{AK_e\Delta p}{\mu l_0}, \tag{5.1}$$

where $Q$ is gas flow (m$^3$ s$^{-1}$); $A$ is seepage cross-sectional area (nm$^2$); $\Delta$p is pressure loss (Pa); $K_e$ is the gas permeability (mD) and $l_0$ is the sample length (nm).

The cross-sectional area of penetration can be expressed as [27]

$$A = \frac{\pi}{4}\frac{D}{2-D}\frac{1-\phi}{\phi}\lambda_{\max}^2, \tag{5.2}$$

where $\lambda_{\max}$ is maximum pore diameter (nm) and $\phi$ is porosity.

The idealized porous media gas flow rate can be expressed as [6]

$$Q = \frac{\pi\lambda_{\max}^4}{128\mu}(1+\delta Kn)\left(1+\frac{4Kn}{1-bKn}\right)\frac{\Delta p}{l_t}, \tag{5.3}$$

where $Q$ is flow (m$^3$ s$^{-1}$); $\delta$ is the thinness factor (dimensionless); $b$ is the gas slip factor and $l_t$ is the thin tube diameter, (nm).

The gas slippage factor can be expressed as [28]

$$b = \frac{4(2-D)^{1/2}}{D^{1/2}(1-\phi)^{1/2}}\frac{\mu}{\lambda_{\max}}\sqrt{\frac{\pi RT}{2M}}. \tag{5.4}$$

The thinning factor can be expressed as [6]

$$\delta = \frac{128}{15\pi^2}\arctan(4Kn^{0.4}). \tag{5.5}$$

Combining equations (4.7)–(5.4) can obtain gas permeability

$$\left.\begin{aligned}K_e &= \frac{\lambda_{\max}^2\phi(1-\phi)\,D}{32\cdot(2-D)\tau}\ \left(1+\frac{128}{15\pi^2}\arctan(4Kn^{0.4})Kn\right)\\ &\times\left(1+\frac{4Kn}{1-(4(2-D)^{1/2}/D^{1/2}(1-\phi)^{1/2})(\mu/\lambda_{\max})(\sqrt{\pi RT/2M})Kn}\right)\frac{l_0}{l_t}.\end{aligned}\right\} \tag{5.6}$$

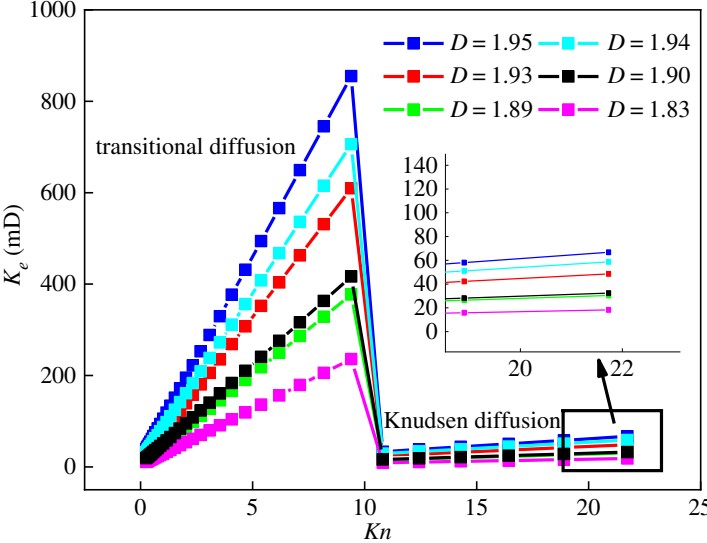

**Figure 8.** The law of $K_e$ variation with Knusen's number under different fractal dimensions.

According to the definition of tortuosity, $\tau = l_0/l_t$, equation (5.6) was substituted into the following [12]:

$$
\left.
\begin{aligned}
K_e = \frac{\lambda_{\max}^2 \phi(1-\phi)\,D}{32\cdot(2-D)\tau}\left(1 + \frac{128}{15\pi^2}\arctan(4Kn^{0.4})Kn\right)\\[2mm]
\times\left(1 + \frac{4Kn}{1 - \dfrac{4(2-D)^{1/2}}{D^{1/2}(1-\phi)^{1/2}}\dfrac{\mu}{\lambda_{\max}}\sqrt{\dfrac{\pi RT}{2M}}Kn}\right).
\end{aligned}
\right\}
\tag{5.7}
$$

The calculation expression of tortuosity of porous media is [29]

$$
\tau = \frac{1}{2}\left[1 + \frac{\sqrt{1-\varphi}}{2} + \sqrt{1-\varphi}\sqrt{\left(\frac{1}{\sqrt{1-\varphi}}-1\right)^2 + \frac{1}{4}}\middle/\left(1-\sqrt{1-\varphi}\right)\right].
\tag{5.8}
$$

## 5.2. Analysis of factors affecting gas flow

Equation (5.7) shows that gas permeability was affected by the $Kn$ number, fractal dimension and porosity. When $T = 323.15$ K and $p = 3$ MPa, the change rule of permeability with fractal dimension is shown in figure 8, and the change rule with porosity is shown in figure 9. It can be seen from figures 8 and 9 that:

(i) In the transitional diffusion zone, the permeability increased with an increase in $Kn$. When $Kn > 10$, the gas diffusion mode changed to Knudsen diffusion and the permeability dropped sharply. This is because the maximum pore diameter of shale in the transition diffusion zone was larger, which in turn increased the cross-sectional area of seepage and the effective pore diameter, [12] which is conducive to the flow of gas. In the Knudsen diffusion zone, the maximum pore diameter of shale decreased sharply, the seepage cross-sectional area and effective pore diameter were reduced, the mean free path of gas molecules was much larger than the pore diameter, and the restraint effect of the gas flow was enhanced. The combined effect increases the resistance to gas flow, resulting in a sharp drop in permeability. In the transition diffusion zone and Knudsen diffusion zone, the permeability increased with the increase in $Kn$, which is consistent with previous research [11,28,30]. The permeability increased with the increase in the fractal dimension, mainly because it gradually increased with the increase in the fractal dimension (figure 10), which in turn strengthened the gas slippage effect and promotes gas migration, [31] increasing permeability.

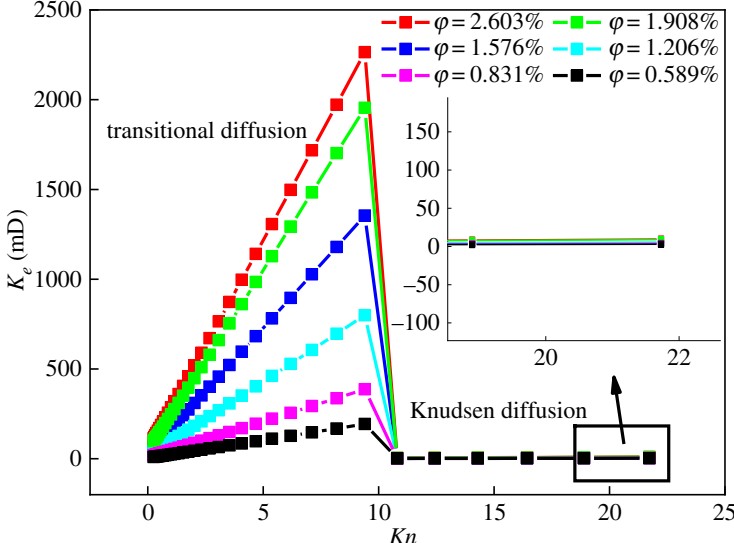

**Figure 9.** Variation of $K_e$ with $Kn$ under different $\varphi$ and $\tau$ conditions.

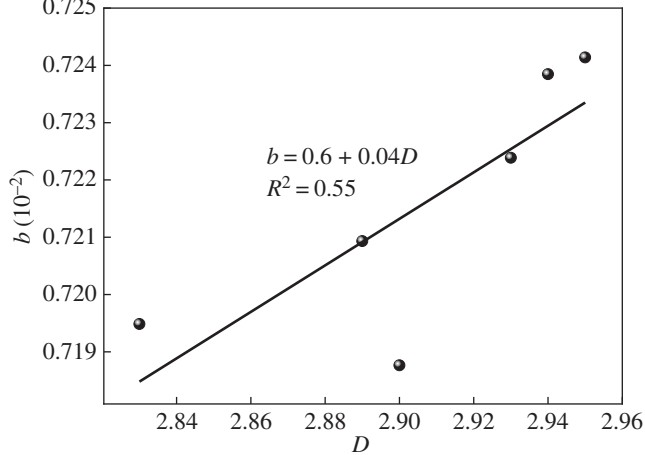

**Figure 10.** Variation of the slip factor with fractal dimension.

(ii) In the transition diffusion zone and Knudsen diffusion zone, the permeability increased with the increase in porosity. The reason is that the greater the porosity, the less resistance to gas flow and the greater the permeability. In the Knudsen diffusion zone, porosity had a small effect on permeability. This may be because, in the Knudsen diffusion zone, micro-nano pores account for more, the gas flow was strongly bound by the pores, and the contribution rate to permeability was small. At the same time, it was affected by amorphous pores such as 'ink bottle' (i.e. shaped pores with poor pore connectivity in the shale), making the increase in permeability insignificant. Compared with the Knudsen diffusion zone, the transition diffusion zone had more mesopores and macropores. Therefore, porosity had a greater influence on permeability.

# 6. Conclusion

In this study, six shale samples were selected from the Niutitang Formation in Guizhou to study the fractal characteristics of the shale pore structure and its influence on seepage. The pore structure characteristics of shale were analysed, the fractal dimensions of the shale pores were calculated using the FHH model, and the functional relationships between the fractal dimension, TOC content and BET surface area were established. The permeability of shale was calculated, and the influence of the

fractal dimension and porosity in different diffusion zones on shale permeability was discussed. The specific conclusions are as follows:

(i) The shale in the Niutitang Formation, Guizhou, has micropores that are generally developed, with larger specific surface areas, smaller pore volumes and average pore diameters.

(ii) The fractal characteristics of the pore surface of the shale in the Niutitang Formation in Guizhou were more obvious. The fractal dimension had a quadratic relationship with the BET specific surface area, and the average pore size and TOC content decreased with an increase in the fractal dimension.

(iii) The diffusion model of shale of the Niutitang Formation in Guizhou was mainly transitional diffusion. The Knudsen number was positively correlated with temperature and decreased with the increase in gas pressure and pore size. The effective diffusion coefficient decreased as the fractal dimension increases.

(iv) Permeability increased with increases in $Kn$, porosity, and the fractal dimension. In the Knudsen area, $\varphi$ had little effect on permeability.

Data accessibility. This article has no additional data.

Authors' contributions. X.L. was involved in data curation and writing the original draft. S.W. was involved in supervision, conceptualization and methodology. H.X. was involved in writing the view and editing. Z.S. was involved in software and validation. L.C. was involved in literature research, manuscript revision and resources.

Competing interests. We declare we have no competing interests.

Funding. This work was supported by the National Natural Science Foundation of China (grant no. 51874107), the Major Applied Basic Research Project of Guizhou Province (grant no. JZ2014-2005) and the Science and Technology Funding Projects of Guizhou Province (grant no. 2018-5781).

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
