## [Peer Review File · Royal Society Open Science]

Review History

RSOS-202271.R0 (Original submission)

Review form: Reviewer 1

Is the manuscript scientifically sound in its present form?

Yes

Are the interpretations and conclusions justified by the results?

Yes

Is the language acceptable?

Yes

Do you have any ethical concerns with this paper?

No

Have you any concerns about statistical analyses in this paper?

No

Recommendation?

Accept with minor revision (please list in comments)

Comments to the Author(s)

This is well written and interesting paper.

See comments below that will improve the readability and consistency of the paper.

domestic and foreign experts -> international research groups

Experimental -> Materials and Methods

was

buried at a depth of approximately 2,500 m, that from Well Tianma 1 was buried at approximately 1,500 m, and that from Well Dafang 1 was buried at approximately 1,000 m. A map of the study area location is shown in Figure 1.

Comment: we dont say "was buried at a depth" -> at a depth of ...

Table 1 X-ray diffraction analysis result of black shale/%. -> what does /% mean. Pleare remove. Also change result to results

Table 2 Porosity nuclear magnetic resonance (NMR) test results -> remove (NMR) from table2 caption

pore size distr

ibution was mainly concentrated -> sentence is broken in pdf

BET specific surface area /S (m²/g) Average pore size /d (nm) Total pore volume per unit mass (cm³/)

remove /S (i am guessing this is the symbol for specific area. If you want to define it, define it in the text in ())

Remove /d .. same as above (cm³/) -> (cm³)

Table 4 Calculation results of pore fractal... -> Calculated pore fractal dimension for each sample.

where k is Boltzmann's constant, 1.38×10^{-23} J/K; d₀ is the gas molecule diameter, methane is 0.38 nm; T is the absolute temperature, K; and p is the gas pressure (MPa). -> please report the units in (). Also K should not be in italics

Pleaca check equation 8, 9 etc for units to be reported in () instead of ,

In equation 10, units for C or DA_{eff} are not given

in eqn 13, nm should be in ()

Previous studies divides the gas diffusion modes into: Kn > 10 indicates Knudsen diffusion; 0.1 < Kn < 10 indicates transition diffusion; 0.01 < Kn < 0.1 indicates Fick diffusion; 0.001 < Kn < 0.01 indicates slip flow; and Kn < 0.001 indicates Darcy flow.

please change as

Previous studies divide the gas diffusion modes into different ranges based on the value of Kn:

(a) $Kn > 10$

which indicates Knudsen diffusion; (b) $0.1 < Kn < 10$ which indicates transition diffusion; (c) $0.01 < Kn < 0.1$ which indicates Fick diffusion; (d) $0.001 < Kn < 0.01$ which indicates slip flow; and (e) $Kn < 0.001$ which indicates Darcy flow.

Fig 1: of-study area -> of study area

Fig 2: diameter/nm -> diameter (nm)

Fig 3. Relative pressure/ -> Relative pressure Adsorption capacity/ -> Adsorption capacity

Fig 4. Make sure all x-axis captions look the same

Fig 5 TOC/% -> TOC (%)

Fig 6. d/nm -> d (nm)

Fig 7. put units in () and delete "/"

Fig 8, 9: put units of K_e in () and delete "/"

Fig 10: put units of b in () and delete "/"

Decision letter (RSOS-202271.R0)

Dear Dr Wang

On behalf of the Editors, we are pleased to inform you that your Manuscript RSOS-202271 "Fractal characteristics of shale pore structure and its influence on seepage flow" has been accepted for publication in Royal Society Open Science subject to minor revision in accordance with the referees' reports. Please find the referees' comments along with any feedback from the Editors below my signature. [There was only one referee for this paper, but their comments are sufficiently clear and positive that I believe it is fair to make a decision without seeking a second report.]

Please submit your revised manuscript and required files (see below) no later than 7 days from today's (ie 15-Apr-2021) date. Note: the ScholarOne system will 'lock' if submission of the revision is attempted 7 or more days after the deadline. If you do not think you will be able to meet this deadline please contact the editorial office immediately.

on behalf of Dr Philip Benson (Associate Editor) and Peter Haynes (Subject Editor)
openscience@royalsociety.org

Reviewer comments to Author:
 Reviewer: 1

Comments to the Author(s)
 This is well written and interesting paper.

See comments below that will improve the readability and consistency of the paper.

domestic and foreign experts -> international research groups

Experimental -> Materials and Methods

Was buried at a depth of approximately 2,500 m, that from Well Tianma 1 was buried at approximately 1,500 m, and that from Well Dafang 1 was buried at approximately 1,000 m. A map of the study area location is shown in Figure 1.

Comment: we dont say "was buried at a depth" -> at a depth of ...

Table 1 X-ray diffraction analysis result of black shale/%. -> what does /% mean. Pleare remove. Also change result to results

Table 2 Porosity nuclear magnetic resonance (NMR) test results -> remove (NMR) from table2 caption

pore size distr
 istribution was mainly concentrated -> sentence is broken in pdf

BET specific surface area /S (m²/g) Average pore size /d (nm) Total pore volume per unit mass (cm³/)

remove /S (i am guessing this is the symbol for specific area. If you want to define it, define it in the text in ())

Remove /d .. same as above
 (cm³/) -> (cm³)

Table 4 Calculation results of pore fractal... -> Calculated pore fractal dimension for each sample.

where k is Boltzmann's constant, 1.38×10^{-23} J/K; d_0 is the gas molecule diameter, methane is 0.38 nm; T is the absolute temperature, K; and p is the gas pressure (MPa). -> please report the units in (). Also K should not be in italics

Please check equation 8, 9 etc for units to be reported in () instead of ,

In equation 10, units for C or D_{Aeff} are not given

in eqn 13, nm should be in ()

Previous studies divide the gas diffusion modes into: $Kn > 10$ indicates Knudsen diffusion; $0.1 < Kn < 10$ indicates transition diffusion; $0.01 < Kn < 0.1$ indicates Fick diffusion; $0.001 < Kn < 0.01$ indicates slip flow; and $Kn < 0.001$ indicates Darcy flow.

please change as

Previous studies divide the gas diffusion modes into different ranges based on the value of Kn :

(a) $Kn > 10$

which indicates Knudsen diffusion; (b) $0.1 < Kn < 10$ which indicates transition diffusion; (c) $0.01 < Kn < 0.1$ which indicates Fick diffusion; (d) $0.001 < Kn < 0.01$ which indicates slip flow; and (e) $Kn < 0.001$ which indicates Darcy flow.

Fig 1: of-study area -> of study area

Fig 2: diameter/nm -> diameter (nm)

Fig 3. Relative pressure/ -> Relative pressure Adsorption capacity/ -> Adsorption capacity

Fig 4. Make sure all x-axis captions look the same

Fig 5 TOC/% -> TOC (%)

Fig 6. d/nm -> d (nm)

Fig 7. put units in () and delete "/"

Fig 8, 9: put units of K_e in () and delete "/"

Fig 10: put units of b in () and delete "/"

===PREPARING YOUR MANUSCRIPT===

===PREPARING YOUR REVISION IN SCHOLARONE===

-- Ensure that your data access statement meets the requirements at <https://royalsociety.org/journals/authors/author-guidelines/#data>. You should ensure that you cite the dataset in your reference list. If you have deposited data etc in the Dryad repository, please only include the 'For publication' link at this stage. You should remove the 'For review' link.

-- If you have uploaded ESM files, please ensure you follow the guidance at <https://royalsociety.org/journals/authors/author-guidelines/#supplementary-material> to include a suitable title and informative caption. An example of appropriate titling and captioning may be found at https://figshare.com/articles/Table_S2_from_Is_there_a_trade-off_between_peak_performance_and_performance_breadth_across_temperatures_for_aerobic_scooping_in_teleost_fishes_/3843624.

Author's Response to Decision Letter for (RSOS-202271.R0)

See Appendix A.

Decision letter (RSOS-202271.R1)

Dear Sir wang,

It is a pleasure to accept your manuscript entitled "Fractal characteristics of shale pore structure and its influence on seepage flow" in its current form for publication in Royal Society Open Science.

on behalf of Dr Philip Benson (Associate Editor) and Peter Haynes (Subject Editor)
openscience@royalsociety.org

Appendix A

Manuscript revision instructions

Dear editor:

We have substantially revised our manuscript after reading the comments provided by the reviewers. We would like to thank the reviewers and the editor for the positive and constructive comments and suggestions. We employed an English-language editing service, Editage, to polish our wording. Certification is attached.

Answer to reviewers:

Comment 1: domestic and foreign experts -> international research groups

Response 1: We have fixed. See revised manuscript.

Comment 2: Experimental -> Materials and Methods:

Response 2: We have fixed. See revised manuscript.

Comment 3: was buried at a depth of approximately 2,500 m, that from Well Tianma 1 was buried at approximately 1,500 m, and that from Well Dafang 1 was buried at approximately 1,000 m. A map of the study area location is shown in Figure 1. Comment: we dont say "was buried at a depth" -> at a depth of ...

Response 3: We have fixed. See revised manuscript.

Comment 4: Table 1 X-ray diffraction analysis result of black shale/%. -> what does % mean. Pleare remove. Also change result to results.

Response 4: We have fixed. See revised manuscript.

Comment 5: Table 2 Porosity nuclear magnetic resonance (NMR) test results -> remove (NMR) from table2 caption.

Response 5: We have Removed (NMR). See revised manuscript.

Comment 6: pore size distribution was mainly concentrated -> sentence is broken in pdf.

Response 6: We have fixed. See revised manuscript.

Comment 7: BET specific surface area /S (m²/g)

Average pore size /d (nm)

Total pore volume per unit mass (cm³/g)

remove /S (i am guessing this is the symbol for specific area. If you want to define it, define it in the text in ()

Remove /d .. same as above

(cm³/g) -> (cm³)

Response 7: We have fixed. See revised manuscript.

Comment 8: Table 4 Calculation results of pore fractal... -> Calculated pore fractal dimension for each sample.

Response 8: We have fixed. See revised manuscript.

Comment 9: where k is Boltzmann's constant, 1.38×10^{-23} J/K; d_0 is the gas molecule diameter, methane is 0.38 nm; T is the absolute temperature, K; and p is the gas pressure (MPa). -> please report the units in (). Also K should not be in italics

Please check equation 8, 9 etc for units to be reported in () instead of ,

In equation 10, units for C or DA_{eff} are not given

in eqn 13, nm should be in ()

Response 9: We have fixed. See revised manuscript.

Comment 10: Previous studies divide the gas diffusion modes into: $Kn > 10$ indicates Knudsen diffusion; $0.1 < Kn < 10$ indicates transition diffusion; $0.01 < Kn < 0.1$ indicates Fick diffusion; $0.001 < Kn < 0.01$ indicates slip flow; and $Kn < 0.001$ indicates Darcy flow.

please change as Previous studies divide the gas diffusion modes into different ranges based on the value of Kn : (a) $Kn > 10$ which indicates Knudsen diffusion; (b) $0.1 < Kn < 10$ which indicates transition diffusion; (c) $0.01 < Kn < 0.1$ which indicates Fick diffusion; (d) $0.001 < Kn < 0.01$ which indicates slip flow; and (e) $Kn < 0.001$ which indicates Darcy flow.

Response 10: We have fixed. See revised manuscript.

Comment 11: Fig 1: of-study area -> of study area

Fig 2: diameter/nm -> diameter (nm)

Fig 3. Relative pressure/ -> Relative pressure Adsorption capacity/ -> Adsorption capacity

Fig 4. Make sure all x-axis captions look the same

Fig 5 TOC/% -> TOC (%)

Fig 6. d/nm -> d (nm)

Fig 7. put units in () and delete "/"

Fig 8, 9: put units of K_e in () and delete "/"

Fig 10: put units of b in () and delete "/"

Response 11: We have fixed. See revised manuscript.